# PaCEr: Network Embedding From Positional to Structural

## ABSTRACT

Network embedding plays an important role in a variety of social network applications. Existing network embedding methods, explicitly or implicitly, can be categorized into positional embedding (PE) methods or structural embedding (SE) methods. Specifically, PE methods encode the positional information and obtain similar embeddings for adjacent/close nodes, while SE methods aim to learn identical representations for nodes with the same local structural patterns, even if the two nodes are far away from each other. The disparate designs of the two types of methods lead to an apparent dilemma in that no embedding could *perfectly* capture both positional and structural information. In this paper, we seek to demystify the underlying relationship between positional embedding and structural embedding. We first point out that the positional embedding can produce the structural embedding with simple transformations, while the opposite direction cannot hold. Based on this finding, a novel network embedding model (PaCEr) is proposed, which optimizes the positional embedding with the help of random walk with restart (RWR) proximity distribution, and such positional embedding is then used to seamlessly obtain the structural embedding with simple transformations. Furthermore, two variants of PaCEr are proposed to handle node classification task on homophilic and heterophilic graphs. Extensive experiments on 17 datasets show that PaCEr achieves comparable or better performance than the state-of-the-arts.

## KEYWORDS

Positional embedding; Structural embedding; Link prediction; Node classification.

## 1 INTRODUCTION

In the age of big data and AI, networks emerge in a variety of high-impact domains, including item recommendation in e-commerce [49], academic co-author recommendation [43], fake news detection [56] and many more. As a pivotal role in these real-world applications, network embedding based approaches aim to map nodes or the whole graph into low dimensional vectors by encoding the topological and attribute information, and have achieved strong empirical performance. Early works [25, 39] utilize matrix factorization to obtain a low-rank representation of the graph. Inspired by Word2Vec [26], DeepWalk [32] and node2vec [14] propose to perform random walks on graph, which are analogous to the sampled positive context in Word2Vec [26]. Metapath2vec [8] and HIN2Vec [11] extend random walks on homogeneous graph to meta-paths to learn the node representations for heterogeneous graph. Recently, graph convolutional network (GCN) [19] has become a powerful graph representation learning paradigm, which embraces the message-passing mechanism to aggregate features from adjacent nodes. Rooted in this message passing mechanism, more advanced GCN architectures have been proposed to strengthen the node representations. To name a few, GraphSAGE [15] randomly samples nodes from two-hop neighbors for feature aggregation,

| PE methods | | | SE methods | |
|---|---|---|---|---|
| MF based | RW based | | SS based | GCN based |
| EVD [1] NetMF [33] GF [2] | DeepWalk [32] node2vec [14] | | SEGK [28] struct2vec [34] GraphWave [9] | GCN [19] GIN [55] GAT [46] |

**Table 1: Some representative positional embedding methods and structural embedding methods. MF represents matrix factorization, RW is the abbreviation for random walk and SS denotes structural similarity.**

GAT [46] introduces the attention mechanism [45] to calculate the weights of connected nodes and APPNP [21] leverages personalized PageRank [29] to capture the information from distant nodes.

Explicitly or implicitly, existing network embedding methods fall into the following two categories [35, 62], including (1) positional embedding (PE) methods, and (2) structural embedding (SE) methods. Positional embeddings obtained by PE methods are expected to seize the relative distance/position information w.r.t. specific nodes (e.g., direct connection, shortest path distance or proximity score). Typical PE methods includes matrix factorization, DeepWalk, node2vec, methpath2vec, etc.[1] On the other hand, SE methods aim to learn similar embeddings for nodes with similar local structural patterns. Struct2vec [34] and most message-passing based GCNs [58, 59] belong to this category. In Table 1, we list some representative PE and SE methods.

However, given the disparate objectives of PE and SE methods, we inevitably encounter the following dilemma in network embedding: there does not exist an embedding that can *perfectly* capture both positional and structural information simultaneously. For an intuitive illustration, we utilize the example in Figure 1. On the one hand, node $u_0$ and $u_6$ have identical topological structure (i.e., they are isomorphic). Consequently, we can obtain same embeddings for node $u_0$ and $u_6$ according to structural information. On the other hand, from the perspective of positional information, $u_0$ is directly connected to $u_1$, while $u_6$ is far away from $u_1$, which forces node $u_0$ and $u_6$ to have divergent embeddings.

Hence, the following two fundamental questions arise, *Q1: How to resolve the dilemma caused by the disparate targets of positional embedding and structural embedding?* and, furthermore, *Q2: What is the underlying relationship between positional embedding and structural embedding?*

To answer *Q1*, existing studies have made tremendous efforts to integrate positional information and structural information in the proposed algorithms. For instance, PGNN [59] constructs anchor node sets and combines the shortest path distance information with message-passing based GCN architectures. Cui et al. [7] explore various positional features (e.g., eigenvector of graph Laplacian and the output of DeepWalk [32]) and structural features (e.g., degree and pagerank score), and integrate them into GraphSAGE [15]. PEG [48] uses different channels to leverage the original node features and the positional features. However, most, if not all, of

---

[1] According to [33], RW based methods can be regarded as implicit matrix factorization.

**Figure 1: An intuitive example to illustrate the dilemma: $u_0$ and $u_6$ are isomorphic node pair. The structural embedding requires that the corresponding embeddings are the same, i.e., $f(u_0) = f(u_6)$. However, since $u_0$ is connected to $u_1$ but $u_6$ is distant from $u_1$, the embeddings of nodes $u_0$ and $u_6$ should be dissimilar from the perspective of positional embedding. No matter what link prediction function $h(\cdot, \cdot)$ is adopted, we will always have $h(f(u_0), f(u_1)) = h(f(u_6), f(u_1))$, which means that it is difficult to predict the existence of the link between $u_0$ and $u_1$, and the non-existence of the link between $u_1$ and $u_6$.**

these works follow a presumption that positional features and structural features are two *distinct* types of information, and have to be *combined* together. [2]

In this paper, we first thoroughly investigate the relation between positional embedding and structural embedding (*Q2*). To be specific, we do not particularly isolate the two types of embeddings and reveal that positional embedding can be transformed to structural embedding with simple operations, while the opposite direction does not hold. Namely, the information encoded in the structural embedding is *redundant* given *informative* positional embedding, and we are therefore motivated to obtain informative positional embedding and perform appropriate transformations to generate the corresponding structural embedding (*Q1*). The key insight is that each row of the normalized adjacency matrix serves as a distribution vector representing the pairwise node proximity (e.g., $p(u_i \rightarrow u_j)$), and we can obtain the positional embedding and the structural embedding by conducting factorization over the distribution and sorting the distribution, respectively.

By further generalizing the above analysis to a distribution matrix defined by node proximity, we propose a novel network embedding algorithm, PₐCEʀ, which can (1) learn high-quality positional embedding, and (2) utilize the learned positional embedding to obtain a corresponding structural embedding in plain networks.[3] The learned positional embedding and structural embedding can be applied to downstream tasks such as link prediction and node classification. Concretely, we first exploit random walk with restart (RWR) [44] to obtain the proximity distribution, serving as the target distribution to be approximated by the product of positional embeddings. Specially, we prove that sorting the RWR proximity distribution in the ascending/descending order will derive the structural embeddings, which has expressive power that is lower-bounded by the Weisfeiler-Lehman (WL) test [51]. Then, the positional embedding is optimized by minimizing the KL-divergence between its *reconstructed distribution* and the input distribution. Subsequently,

PₐCEʀ sorts the *reconstructed* proximity distribution for each node and yields the structural embedding. In addition, to deal with graphs where nodes have attributes, we present two variants of PₐCEʀ: PₐCEʀ-A targets at homophilic graph where connected nodes tend to own similar attributes/features, while PₐCEʀ-H is designed for heterophilic graph where connected nodes may not have similar attributes/features. Through extensive empirical evaluations on 17 real-world datasets in the following four tasks, including (1) link prediction, (2) structural node classification [7] [4], (3) homophilic node classification, and (4) heterophilic node classification, we corroborate the high quality of the obtained positional and structural embeddings from PₐCEʀ,

To summarize, our contributions are threefold:

- **Theoretical Analysis.** Through analysis on the relation between positional embedding and structural embedding, we find that the positional embedding can actually induce the structural embedding with simple transformations.
- **Novel Algorithms.** Based on the theoretical analysis, we propose a network embedding model PₐCEʀ, which innovatively generates positional embeddings with RWR proximity distribution and utilizes the positional embedding to obtain the structural embedding. In addition, we propose two variants of PₐCEʀ: PₐCEʀ-A and PₐCEʀ-H to solve node classification task on homophilic and heterophilic graphs respectively.
- **Experimental Results.** We conduct extensive experiments on 17 datasets and empirically find that the proposed PₐCEʀ achieves comparable or better performance than the state-of-the-arts in four representative graph learning tasks, which demonstrates the superiority of PₐCEʀ.

## 2 PROBLEM DEFINITION

In this section, we introduce the definitions of positional embedding and structural embedding. We first summarize the main symbols used in this paper. We adopt bold uppercase letters for matrices (e.g., $\mathbf{A}$), bold lowercase letters for vectors (e.g., $\mathbf{v}$), calligraphic letters for sets (e.g., $\mathcal{A}$) and lowercase letters for scalars (e.g., $a$). In addition, we follow the convention in Matlab to represent the $u$-th row of matrix $\mathbf{A}$ as $\mathbf{A}(u, :)$, the $v$-th column as $\mathbf{A}(:, v)$, the $(u, v)$-th entry as $\mathbf{A}(u, v)$ and the transpose of matrix $\mathbf{A}$ as $\mathbf{A}^\top$.

Next, we give the formal definitions of *node permutation*, *node isomorphism* and *permutation invariant function*, followed by the definitions of structural embedding and positional embedding. We mainly consider an undirected graph without node attributes, which can be represented as $G = (\mathcal{V}, \mathcal{E}, \mathbf{A})$, where $\mathcal{V}$ is the node set, $\mathcal{E} \subset \mathcal{V} \times \mathcal{V}$ is the edge set, and $\mathbf{A}$ is the adjacency matrix. [41, 61] define *node permutation*, *node isomorphism* and *permutation invariant function* as follows:

**DEFINITION 1.** *Node Permutation.* Let $\pi$ represent the node index permutation, which is a bijective mapping from $\mathcal{V}$ to $\mathcal{V}$. All possible permutations form the permutation group $\prod_n$, where $n = |\mathcal{V}|$. If $\pi$ acts on a node subset $\mathcal{S}$ of $\mathcal{V}$, it is denoted as $\pi(\mathcal{S}) = \{\pi(u_i) | u_i \in \mathcal{S}\}$.

**DEFINITION 2.** *Node Isomorphism.* For two nodes $u_i$ and $u_j$ belonging to $\mathcal{V}$, given a non-negative integer hop number $k \in \mathcal{N}$, the

---

[2]In [41], the authors prove that positional embedding is equivalent to *multi-node set* level structural embedding, which refers to the structural embedding of a multi-node set and is different from the *single-node* level structural embedding we want to study in this paper.

[3]A plain network refers to a graph whose nodes and edges do not possess attributes.

[4]Structural node classification targets at classifying nodes according to their local structural patterns.

$k$-hop subgraphs starting from $u_i$ and $u_j$ are denoted as $G_{u_i}^{(k)} = (\mathcal{S}_{u_i}^{(k)}, \mathcal{E}_{u_i}^{(k)}, \mathbf{A}_{u_i}^{(k)})$ and $G_{u_j}^{(k)} = (\mathcal{S}_{u_j}^{(k)}, \mathcal{E}_{u_j}^{(k)}, \mathbf{A}_{u_j}^{(k)})$. We call that nodes $u_i$ and $u_j$ are $k$-hop isomorphic (otherwise $k$-hop non-isomorphic), if $\exists \pi \in \prod_n$ such that $\mathcal{S}_{u_i}^{(k)} = \pi(\mathcal{S}_{u_j}^{(k)})$ and $\mathbf{A}_{u_i}^{(k)} = \pi(\mathbf{A}_{u_j}^{(k)})$. If $\forall k \in \mathcal{N}$, $u_i$ and $u_j$ are $k$-hop isomorphic, we call them isomorphic.

DEFINITION 3. **Permutation Invariant Function.** A function $f$ defined on $\mathcal{S}_{u_i}$ ($i = 1, 2, \ldots, n$) is permutation invariant if $\forall \pi \in \prod_n$, $f(\mathcal{S}_{u_i}, \mathbf{A}) = f(\pi(\mathcal{S}_{u_i}), \pi(\mathbf{A}))$.

With the above definitions of *node permutation*, *node isomorphism* and *permutation invariant function*, we give the formal definition of the structural embedding as the following:

DEFINITION 4. **Structural Embedding.** The $k$-hop structural embedding of node $u_i$ can be defined as a function $f^{(k)}(\cdot)$ on $\mathcal{S}_{u_i}^{(k)}$, where $f$ is permutation invariant. When $k \to \infty$, we use $f(u_i)$ to denote the structural embedding of node $u_i$ in the entire graph $G$.

From the definition of *structural embedding*, we can see that the $k$-hop structural embeddings of two $k$-hop isomorphic nodes $u_i$ and $u_j$ should be identical (i.e., $f^{(k)}(u_i) = f^{(k)}(u_j)$). In addition, the structural embedding should not be affected by the node indexing because it is obtained by a permutation invariant function on $\mathcal{S}_{u_i}^{(k)}$. We give the definition of positional embedding as follows [22]:

DEFINITION 5. **Positional Embedding.** The positional embedding of node $u_i$ can be denoted as $g(u_i)$, which is used to encode the positional information $\phi(l_{u_i, u_j})$ from $u_i$ to any $u_j \in \mathcal{V}$. Here $\phi(\cdot)$ is an encoding function and $l_{u_i, u_j} = (\mathbf{W}(u_i, u_j), \ldots, \mathbf{W}^k(u_i, u_j), \ldots)$ is the positional information, where $\mathbf{W} = \mathbf{D}^{-1}\mathbf{A}$ is one hop random walk matrix and $\mathbf{D}$ is the degree matrix.

According to [22], $\phi(\cdot)$ can be implemented with various encoding strategies. For example, if the output of $\phi$ is the index $k$ of the first non-zero element in $l_{u_i, u_j}$, $\phi(\cdot)$ represents the shortest path distance, while it corresponds to a $k$-step random walk if $\phi(l_{u_i, u_j}) = \mathbf{W}^k(u_i, u_j)$.

# 3 THE PACER MODEL

In this section, we reveal the underlying relationship between positional embedding and structural embedding, and present the proposed model PaCEr. We start with the observation and our analysis of row-normalized adjacency matrix, which inspires us to come up with the key idea that the positional embedding can generate a pairwise proximity distribution, and sorting the proximity distribution for each node can produce a structural embedding (Subsection 3.1). Motivated by this idea, we first exploit random walk with restart (RWR) [44] to acquire the target proximity distribution matrix, and prove that the structural embedding can be obtained by simply sorting each row of the proximity distribution matrix. Particularly, the vanilla structural embeddings obtained by RWR have expressive power not worse than the WL-test [51]. Based on this target proximity distribution, we adopt the KL-divergence to optimize the positional embedding (Subsection 3.2). Next, we demonstrate that via some straightforward transformations (i.e., multiplication and sorting), the positional embedding can be successfully transformed into the corresponding structural embedding in PaCEr with the

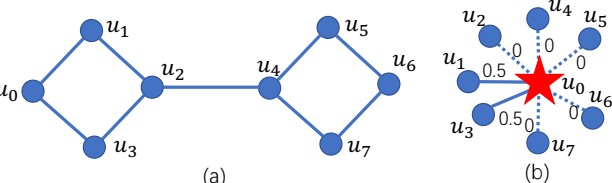

(a)        (b)

**Figure 2: The left graph is the toy example from Figure 1, where $u_0$ has links connected to $u_1$ and $u_3$ in the spatial domain. The right graph is the augmented graph constructed by $\hat{\mathbf{A}}$, where $u_0$ has links connected to the remaining nodes (e.g., $u_5$) with weight 0 marked with dashed lines.**

help of the intermediate reconstructed proximity distribution. In addition, we propose two variants of PaCEr: PaCEr-A and PaCEr-H, which generalize the *structural embedding* to *attributed structural embedding* on both homophilic and heterophilic graphs (Subsection 3.3). Finally, we analyze the complexity of PaCEr and have a brief discussion about the relationship between PaCEr and existing works from different research lines, which demonstrates that PaCEr could act as a *bridge* between these different research lines (Subsection 3.4).

## 3.1 Analysis and Key Idea

In this subsection, we present the key insight of the proposed PaCEr: *positional embedding can be transformed into structural embedding by viewing each row of the row-normalized adjacency matrix as a proximity distribution.*

Given the adjacency matrix of a graph (i.e., $\mathbf{A}$), the row-normalized adjacency matrix ($\hat{\mathbf{A}}$) is calculated by dividing each entry in the original $\mathbf{A}$ by the degree of corresponding node: $\hat{\mathbf{A}}(u_i, u_j) = \frac{\mathbf{A}(u_i, u_j)}{d_{u_i}}$, where $d_{u_i}$ is the degree of node $u_i$. The row-normalized adjacency matrix is widely used in many powerful graph algorithms like GCN [19] and PageRank [29]. Usually, $\hat{\mathbf{A}}$ is interpreted in the spatial domain. For example, $\hat{\mathbf{A}}$ functions as one hop random walk in DeepWalk [32] or one hop message-passing in GCN [19] for neighbor nodes.

Here, we interpret the row-normalized adjacency matrix from the *proximity distribution* perspective rather than the spatial domain. Taking the graph in Figure 1 as an example, $u_0$ is linked to $u_1$ and $u_3$, and $\hat{\mathbf{A}}(u_0, :) = [0, \frac{1}{2}, 0, \frac{1}{2}, 0, 0, 0, 0]$. By regarding $\hat{\mathbf{A}}(u_0, :) = [0, \frac{1}{2}, 0, \frac{1}{2}, 0, 0, 0, 0]$ as a proximity distribution, we can construct a new augmented graph for $u_0$ as shown in Figure 2(b). The difference between the augmented graph in Figure 2(b) and the original graph in Figure 2(a) is that node $u_0$ is *connected to* nodes $u_2, u_4, u_5, u_6, u_7$ with 0 weighted edges in the augmented graph, while in Figure 2(a), $u_0$ has no edges linked to these nodes in the spatial domain.

With the augmented graph and proximity distribution, we demystify the relation between the positional embedding and the structural embedding. First, according to [20, 33, 41], the key idea of most PE methods, including random-walk based methods [33], graph auto-encoder (GAE) [20] and eigen-decomposition [41], is to conduct matrix factorization on specific matrices. Among these matrices, the row-normalized adjacency matrix (i.e., $\hat{\mathbf{A}}$) is a widely used one. From the view of proximity distribution, conducting matrix factorization over $\hat{\mathbf{A}}$ and optimizing the positional embedding is equivalent to approximating the proximity distribution, e.g.,

minimizing $\|h(g(u_i), g(u_j)) - p(u_i \rightarrow u_j)\|$, where $g(u_i), g(u_j)$ represent positional embeddings, $h(\cdot)$ is the pairwise function over node embeddings such as dot product, and $p(u_i \rightarrow u_j) = \hat{\mathbf{A}}(u_i, u_j)$ is the proximity score (probability). We hence observe that the positional embedding is used to *reconstruct* the proximity distribution matrix (i.e., $\hat{\mathbf{A}}$) with the help of pairwise function $h(\cdot, \cdot)$.

Furthermore, we investigate the relation between the proximity distribution and structural embedding. By simply sorting each row of $\hat{\mathbf{A}}$, we notice that (1) node $u_1$ and node $u_7$ have identical sorted vectors, i.e., $[\frac{1}{2}, \frac{1}{2}, 0, 0, 0, 0, 0, 0]$, and (2) the sorted vectors for node $u_2$ and node $u_4$ are the same, i.e., $[\frac{1}{3}, \frac{1}{3}, \frac{1}{3}, 0, 0, 0, 0, 0]$. According to Definition 2 of node isomorphism, node pair $(u_1, u_7)$ and $(u_2, u_4)$ are 1-hop isomorphic respectively, i.e., owning the same degree. From this example, we can see that with a *sorting* operation, the proximity distribution can be transformed into a structural embedding, which has the expressive power of testifying whether two nodes are 1-hop isomorphic.

By incorporating the relation between proximity distribution and structural/positional embedding, we introduce the key idea of the proposed PaCEr as follows. In the ideal case, the positional embedding is *sufficiently informative* to *fully reconstruct* the proximity distribution matrix (e.g., $\hat{\mathbf{A}}$) in the first step. In the second step, we can acquire the structural embedding by a *sorting* operation $R(\cdot)$ on the reconstructed proximity distribution. This reveals the relation between positional embedding and structural embedding: we can utilize the positional embedding to produce the structural embedding correspondingly. However, for the opposite direction, the structural embedding cannot generate the positional embedding. For example, $R(\hat{\mathbf{A}}(u_0, :))$ is the structural embedding and it cannot restore the original $\hat{\mathbf{A}}(u_0, :)$, i.e., we are unable to restore the original proximity distribution $[0, \frac{1}{2}, 0, \frac{1}{2}, 0, 0, 0, 0]$ from $[\frac{1}{2}, \frac{1}{2}, 0, 0, 0, 0, 0, 0]$ after permutation.

The relationship between positional embedding and structural embedding can also be explained in an intuitive manner: the row-normalized adjacency matrix $\hat{\mathbf{A}}$ contains all information to learn the positional embedding and the structural embedding. If the positional embedding is *informative* enough to reconstruct $\hat{\mathbf{A}}$, it has no loss on the topological information, while the structural embedding is unable to re-establish $\hat{\mathbf{A}}$ and bears topological information loss. This also highlights the fundamental difference between PaCEr and previous works: we point out that all the information captured by the structural embedding is implicitly contained by an *informative* positional embedding. This suggests that it might be unnecessary to employ redundant channels or modules to incorporate the structural information into the positional embedding [7, 48]. Therefore, the proposed PaCEr instead focuses on training an *informative* positional embedding and construct an intermediate proximity distribution matrix, based on which the structural embedding can be obtained.

Nevertheless, the row-normalized adjacency matrix $\hat{\mathbf{A}}$ is not an ideal choice of the proximity distribution matrix for the positional embedding to approximate due to the following reasons. First, $\hat{\mathbf{A}}$ is too coarse to capture positional information because $p(u_i \rightarrow u_j) \neq 0$ only when $u_i$ and $u_j$ are directly connected. This means that nodes outside 1 hop distance are treated equally with $p(u_i \rightarrow u_j) = 0$ (e.g., $p(u_1 \rightarrow u_3) = p(u_1 \rightarrow u_7) = 0$). Second, the

structural embedding built upon $\hat{\mathbf{A}}$ merely possesses the expressive power to test whether two nodes are 1-hop isomorphic, which is insufficient for $k$-hop isomorphism problem, where $k > 1$. How can we find a proximity distribution matrix which is capable of (1) accurately documenting the positional information, and (2) obtaining better structural expressive power as the target distribution to be approximated?

## 3.2 Positional Embedding Module

In this subsection, we introduce the positional embedding module of PaCEr. We choose the transpose of random walk with restart (RWR) [44] to build the initial proximity distribution matrix: $\mathbf{r}_{u_i} = c\hat{\mathbf{A}}^\top \mathbf{r}_{u_i} + (1-c)\mathbf{e}_{u_i}$, where $\mathbf{r}_{u_i}$ denotes the node proximity vector for node $u_i$, $1 - c$ is the restart probability and $\mathbf{e}_{u_i}$ represents a one-hot starting vector with the $u_i$-th element equal to 1 and all the remaining elements equal to 0. From the above equation, we can obtain the closed-form solution of $\mathbf{r}_{u_i}$ as follows: $\mathbf{r}_{u_i} = (1-c)\sum_{k=0}^{\infty}(c\hat{\mathbf{A}}^\top)^k\mathbf{e}_{u_i}$, where $k$ is the hop number. We can also derive the matrix form as $\mathbf{R} = (1-c)\sum_{k=0}^{\infty}(c\hat{\mathbf{A}}^\top)^k\mathbf{I}$, where each column of $\mathbf{R}$ represents the RWR vector, e.g., $\mathbf{r}_{u_i}$. We then choose the transpose of the RWR matrix $\mathbf{R}^\top$ as the target proximity distribution matrix.

Compared to the row-normalized matrix $\hat{\mathbf{A}}$, $\mathbf{R}^\top$ enjoys the following two strengths in building positional embedding and structural embedding. First, we can observe that $\mathbf{r}_{u_i}$ can seize the distance/positional information for all nodes within $k$ hops ($k \in \mathcal{N}$) rather than only one hop neighborhood as in $\hat{\mathbf{A}}$. In addition, $\mathbf{R}^\top$ extends the proximity in $\hat{\mathbf{A}}$ from discrete values to continuous values between [0,1], which are more accurate and contain more detailed information regarding node-wise proximity. Second, regarding structural embedding, as mentioned in Subsection 3.1, $\hat{\mathbf{A}}$ possesses the expressive power of testing whether two nodes are 1-hop isomorphic or not. A natural question arises: *how is the expressive power of $R(\mathbf{R}^\top(u_i, :))$?* To answer this, we have the following proposition regarding the expressive power of $R(\mathbf{R}^\top(u_i, :))$:

PROPOSITION 1. *$R(\mathbf{R}^\top(u_i, :))$ is permutation invariant and its expressive power is not worse than the expressive power of 1-WL test.*

PROOF. See Appendix 7.1. □

Similar to WL-test [51], $R(\mathbf{R}^\top(u_i, :))$ and $R(\mathbf{R}^\top(u_j, :))$ can also be used to verify whether two nodes $u_i$ and $u_j$ are isomorphic or not. We have the following node isomorphism test algorithm with $R(\mathbf{R}^\top(u_i, :))$ in Algorithm 1. Note that Algorithm 1 can test whether two nodes $u_i$ and $u_j$ are non-isomorphic but can not ensure these are definitely isomorphic, like the WL-test.[5]

Equipped with the two strengths of $\mathbf{R}^\top$, we propose PaCEr for bridging positional embedding and structural embedding, which is composed of two modules (1) positional embedding module and (2) structural embedding module. The overview of PaCEr is presented in Figure 3. Specifically, for the positional embedding module, we first calculate the transpose of the RWR matrix ($\mathbf{R}^\top$). Then, given node $u_i$, we follow [14, 32] and randomly initialize a vector as the positional embedding $g(u_i)$ of node $u_i$ (i.e., $g(u_i) = \mathbf{u}_i$), which is to be optimized by PaCEr. The proximity distribution matrix

---

[5]Since the isomorphism problem is NP hard, the *best output* of *any* method is "possible-isomorphic" under polynomial time complexity.

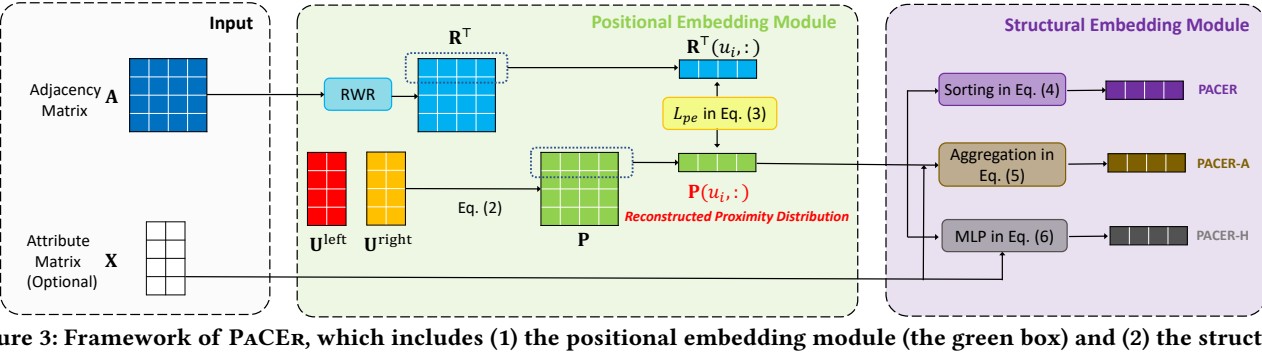

**Figure 3: Framework of PaCEr, which includes (1) the positional embedding module (the green box) and (2) the structural embedding module (the purple box).**

---

**Algorithm 1** Node Isomorphism Test Algorithm by $R(\mathbf{R}^\top(u_i, :))$ and $R(\mathbf{R}^\top(u_j, :))$ .

---

**Input:** (1) The adjacency matrix $\mathbf{A}$ for the graph $G$; (2) the restarting probability $1 - c$; (3) two nodes to be tested: $u_i$ and $u_j$; (4) the maximum iteration (i.e., the maximum hop number) $k_{\max}$.

**Output:** Return "non-isomorphic" and the $k_{\text{first}}$ if $G_{u_i}^{(k)}$ and $G_{u_j}^{(k)}$ are isomorphic for $k < k_{\text{first}}$ but are non-isomorphic when $k = k_{\text{first}}$ or "possible-isomorphic" when $G_{u_i}^{(k_{\max})}$ and $G_{u_j}^{(k_{\max})}$ are isomorphic.

    Calculate the row-normalized adjacency matrix $\hat{\mathbf{A}}$ with $\mathbf{A}$;
    **while** $k \leq k_{\max}$ **do**
        Calculate $\mathbf{R}^\top(u_i, :)$ ($\mathbf{r}_{u_i}^\top$) and $\mathbf{R}^\top(u_j, :)$ ($\mathbf{r}_{u_j}^\top$);
        Sort the proximity score within $\mathbf{R}^\top(u_i, :)$ and $\mathbf{R}^\top(u_i, :)$ in a descending order to obtain $R(\mathbf{R}^\top(u_i, :))$ and $R(\mathbf{R}^\top(u_j, :))$;
        **if** $R(\mathbf{R}^\top(u_i, :)) \neq R(\mathbf{R}^\top(u_i, :))$ **then**
            **return** "non-isomorphic" and $k_{\text{first}} = k$.
        **end if**
        $k = k + 1$;
    **end while**
    **return** "possible-isomorphic".

---

$\mathbf{R}^\top$ is asymmetric (i.e., $\mathbf{R}^\top(u_i, u_j) \neq \mathbf{R}^\top(u_j, u_i)$), hence directly applying the dot product $h(u_i, u_j) = \mathbf{u}_i^\top \mathbf{u}_j$ (one single value) to approximate both $\mathbf{R}^\top(u_i, u_j)$ and $\mathbf{R}^\top(u_j, u_i)$ (two different values) is not feasible. Therefore, we conduct a column-wise split on the positional embedding $\mathbf{u}_i$, which can be divided into the left half $\mathbf{u}_i^{\text{left}}$ and the right half $\mathbf{u}_i^{\text{right}}$:

$$\mathbf{u}_i = \text{CONCAT}(\mathbf{u}_i^{\text{left}}, \mathbf{u}_i^{\text{right}}). \tag{1}$$

Based on Eq. (1), we denote the positional embedding of the graph $G$ as $\mathbf{U} = \text{CONCAT}(\mathbf{U}^{\text{left}}, \mathbf{U}^{\text{right}})$. Thus, the reconstructed proximity distribution $\mathbf{P}(u_i, :)$ can be calculated as:

$$\mathbf{P}(u_i, :) = \text{Softmax}(\mathbf{U}^{\text{left}}(u_i, :)\mathbf{U}^{\text{right}\top}), \tag{2}$$

where $\text{Softmax}(\cdot)$ [5] is utilized to normalize the vector as a probability distribution.

Ultimately, we leverage KL-divergence to optimize the positional embedding by minimizing the distance between the reconstructed proximity distribution and the target proximity distribution as follows:

$$L_{pe} = \sum_{u_i \in \mathcal{V}} \text{KL}(\mathbf{R}^\top(u_i, :), \mathbf{P}(u_i, :)). \tag{3}$$

## 3.3 Structural Embedding Module

In this subsection, we present the structural embedding module of PaCEr. In addition, we propose two variants of PaCEr for attributed graphs where node attributes are available, including PaCEr-A for homophilic graphs and PaCEr-H for heterophilic graphs.

As mentioned in Subsection 3.1, PaCEr is able to learn an *informative* positional embedding, and this positional embedding can be manipulated to generate the corresponding structural embedding. Similar to the analysis on the proximity distribution matrix (i.e., $\mathbf{R}^\top$), PaCEr sorts the reconstructed proximity distribution $\mathbf{P}(u_i, :)$ after $L_{pe}$ converges to obtain the structural embedding for $u_i$ as:

$$f(u_i) = R(\mathbf{P}(u_i, :)), \tag{4}$$

where $f(u_i)$ is the structural embedding of node $u_i$ and $R(\cdot)$ is the sorting operation.

However, the *structural embedding* in Eq. (4) does not encode the node attribute information (i.e., $\mathbf{X} \in \mathbb{R}^{n \times d_x}$), where $d_x$ is the dimension of node attribute $\mathbf{X}(u_i, :)$. To address this, we further propose two variants of PaCEr, named PaCEr-A and PaCEr-H.

**PaCEr-A.** For homophilic graphs where close nodes tend to possess similar attributes and labels, we propose PaCEr-A. We notice that $\mathbf{P}(u_i, u_j)$ measures the proximity score between node $u_i$ and $u_j$, and $\mathbf{P}(u_i, :)$ represents a distribution. Therefore, we can directly employ $\mathbf{P}(u_i, u_j)$ as the probability of propagating feature/attribute from $u_i$ to $u_j$ ($p(u_i \rightarrow u_j)$):

$$f(u_j) = \sum_{u_i} \mathbf{P}(u_i, u_j)\mathbf{X}(u_i, :). \tag{5}$$

We can observe that Eq. (5) is very similar to the one-layer GCN's message-passing on the augmented graph in Figure 2(b). As shown in Figure 2(b), compared to the augmented graph from the row-normalized matrix $\hat{\mathbf{A}}$ where non-zero edge weights only exist for one-hop neighborhood, the graph obtained from the proximity distribution matrix (i.e., $\mathbf{P}(u_i, u_j)$) can capture more detailed information between nodes regardless of connectivity.

**PaCEr-H.** *Heterophilic graphs* refer to networks where topological closeness does not mean positive correlations w.r.t. node attributes and labels between nodes. To tackle this challenge, we propose the second variant of PaCEr: PaCEr-H. Different from PaCEr-A, PaCEr-H does not directly assume that proximity $\mathbf{P}$ and the node attributes are *positively correlated* to each other, and does not employ Eq. (5) to calculate $f(u_i)$. Therefore, PaCEr-H exploits a

neural network architecture, i.e., multilayer perceptron (MLP) [16], to automatically learn such complex correlations. Specifically, we concatenate the topological information (i.e., $\mathbf{P}(u_i, :)$) and the attribute information (i.e., $\mathbf{X}(u_i, :)$) and feed it to the neural network. Mathematically, we compute the structural embedding as follows:

$$f(u_i) = \text{MLP}(\text{CONCAT}(\mathbf{P}(u_i, ; ), \mathbf{X}(u_i, :))). \tag{6}$$

## 3.4 Complexity Analysis and Discussions

In this subsection, we have a brief complexity analysis on PACER and discuss the relation between PACER and existing works from different research lines.

**Complexity Analysis.** The time complexity of the vanilla implementation of PACER can be divided into the following parts: (1) Calculating $\mathbf{R}^{\top}(u_i, :)$ has the complexity of $O(k_{\max} \cdot |\mathcal{E}|)$, where $|\mathcal{E}|$ is the number of edges in the graph and $k_{\max}$ is the maximum iterations or hops; (2) the time complexity for computing the $\mathbf{P}(u_i, :)$ matrix and the KL-divergence between $\mathbf{P}(u_i, :)$ and $\mathbf{R}^{\top}(u_i, :)$ is $O(n \cdot \frac{d_p^2}{4} + n)$, where $d_p$ is the dimension of positional embedding; (3) Sorting $\mathbf{P}(u_i, :)$ needs $O(n \cdot \log(d_s))$, where $d_s$ is the dimension of the structural embedding;[6] (4) Obtaining $f(u_j)$ in Eq. (5) needs $O(n \cdot d_x)$.

**Discussion.** Here, we discuss the connections and the differences between PACER and existing works from the following three aspects. First, PACER primarily focuses on studying the relations between positional embeddings and structural embeddings [7, 41, 48]. Second, if we replace $\mathbf{P}$ with the initial $\mathbf{R}^{\top}$, we observe that PACER-A becomes PPNP in APPNP [13]. Actually, $\mathbf{P}$ can be generalized to different proximity distribution matrices. For example, using $\hat{\mathbf{A}}$ as the proximity distribution matrix leads to the classical GCN [19], and iteratively updating the proximity distribution matrix with attention mechanism results in the key component in GAT [46]. Third, under the assumption that the correlation between the positional information and the attribute is unknown in heterophilic graphs, PACER-H directly uses an MLP to learn such correlation. This is consistent with the prior findings in [24] that simply concatenating the original adjacency matrix $\mathbf{A}$ and the attribute matrix $\mathbf{X}$, and applying an MLP to this concatenation can achieve significant performance on node classification task on heterophilic graphs. All in all, PACER can act as an *intermediate bridge* to connect various lines of existing works.

## 4 EXPERIMENT

In this section, we evaluate the proposed PACER from the following aspects:

- How effective is the positional embedding obtained by PACER? (*Effectiveness of PACER in link prediction*)
- How effective is the structural embedding obtained by PACER? (*Effectiveness of PACER in structural node classification*)
- To what extent can PACER be generalized to attributed graphs? (*Effectiveness of PACER-A and PACER-H in attributed node classification on homophilic graphs and heterophilic graphs*)

---

[6]We usually choose the largest $d_s$ values of $\mathbf{P}(u_i, :)$ as the structrual embedding in implementation.

## 4.1 Experimental Setup

In this subsection, we introduce the datasets, metrics, and baselines used for four tasks: (1) link prediction, (2) structural node classification, (3) homophilic node classification, and (4) heterophilic node classification in our experiments. In addition, we detail the parameter settings of PACER in different tasks.

**Datasets.** In the experiments, we adopt 17 commonly used real-world datasets in total to evaluate the proposed PACER for the above four tasks. The statistics of all datasets are listed in Table 2. The detailed descriptions and splits of the datasets are attached in Appendix 7.2 due to page limit.

| Dataset | Task | Nodes | Edges | Features | Classes |
|---|---|---|---|---|---|
| NS [27] | | 1,589 | 2,742 | - | - |
| Ecoli [40] | | 1,805 | 14,660 | - | - |
| USAir [47] | Link Prediction | 322 | 2,126 | - | - |
| Celegans [50] | | 297 | 2,148 | - | - |
| Citeseer [57] [7] | | 3,327 | 9,104 | - | - |
| Cora [57] | | 2,708 | 10,556 | - | - |
| Brazil [34] | Structural | 131 | 1,038 | - | 4 |
| Europe [34] | Node Classification | 399 | 5,995 | - | 4 |
| USA [34] | | 1,190 | 13,599 | - | 4 |
| Cora [57] | | 2,708 | 10,556 | 1,433 | 7 |
| Citeseer [57] | | 3,327 | 9,104 | 3,703 | 6 |
| Pubmed [57] | | 19,717 | 88,648 | 500 | 3 |
| Computers [38] | Homophilic Node Classification | 13,752 | 491,722 | 767 | 10 |
| Photo [38] | | 7,650 | 238,162 | 745 | 8 |
| CS [38] | | 18,333 | 163,788 | 6,805 | 15 |
| DBLP [4] | | 17,716 | 105,734 | 1,639 | 4 |
| Squirrel [36] | Heterophilic | 5,201 | 396,846 | 2,089 | 5 |
| Actor [31] | Node Classification | 7,600 | 30,019 | 932 | 5 |
| Cornell5 [24] | | 18,660 | 1,581,554 | 4,735 | 2 |

**Table 2: Dataset statistics.**

**Metrics.** For the link prediction task, we use the Area Under the ROC and Precision-Recall Curves (i.e.,AUC-ROC and AUC-PR) as the metrics to evaluate the performance of different methods. For the remaining three tasks (i.e., (1) structural node classification, (2) homophilic node classification and (3) heterophilic node classification), we use the classification accuracy (ACC) as the metric. For all baselines and PACER, we report the average AUC-ROC/AUC-PR/ACC with the standard deviation in 5 runs.

**Baselines.** We compare PACER with 16 baselines. For the link prediction task, we have the following 5 baselines: node2vec [14], VGAE [20], GAT[46], ARGVA [30] and RWBGE [17]. For the structural node classification task, we have these baselines: GraphSAGE [15], GCN [19], Union [23], Intersection [23], GAT [46], Demo-Net [53], and GraphWave [9]. For both homophilic node classifcation and heterophilic node classification, we use the following attributed node classification baselines: GCN [19], SGC [52], APPNP [13], GPRGNN [6], FAGCN [3] and H2GCN [63].

**Parameter Settings.** The dimension for node embeddings in all baselines are set as 128 and we set the dimension of $\mathbf{u}_i^{\text{left}}$ and $\mathbf{u}_i^{\text{right}}$ as 64 (64 + 64 = 128) for a fair comparison. For the positional embedding optimization module, we set the number of epochs as 5,000. The learning rate is set as 0.01. For the structural embedding optimization module, including PACER-A and PACER-H, we set the number of epochs as 2,000 and the learning rate as 0.001. The restart probability in RWR is set as $(1 - c) = 0.15$. All experiments are run on a Tesla-V100 GPU. We will release the code after publication.

| Models | Cora | | Citeseer | | NS | | Ecoli | | USAir | | Celegans | |
|---|---|---|---|---|---|---|---|---|---|---|---|---|
| | AUC-ROC | AUC-PR | AUC-ROC | AUC-PR | AUC-ROC | AUC-PR | AUC-ROC | AUC-PR | AUC-ROC | AUC-PR | AUC-ROC | AUC-PR |
| node2vec | 75.96±1.18 | 82.73±0.69 | 69.63±0.97 | 77.38±0.89 | 86.48±0.76 | **91.36±0.75** | 76.61±0.38 | 77.03±0.48 | 79.28±2.30 | 75.38±3.22 | 79.36±1.14 | 74.27±1.33 |
| VGAE | 78.28±0.53 | 81.07±0.59 | 72.58±0.74 | 78.00±0.40 | 86.45±0.95 | 89.99±0.58 | 90.94±0.22 | 93.30±0.23 | 89.74±1.56 | **90.55±1.56** | 77.78±1.39 | 73.50±2.38 |
| GAT | 76.48±0.77 | 78.90±1.03 | 71.79±1.40 | 76.04±1.18 | 85.89±0.84 | 88.47±1.92 | 90.32±0.55 | 92.73±0.45 | 88.67±1.28 | 89.85±1.48 | 74.16±4.33 | 71.21±1.69 |
| ARGVA | 75.99±0.97 | 78.11±1.14 | 71.96±1.10 | 74.74±0.59 | **88.55±1.12** | 90.32±1.36 | 91.02±0.51 | 93.31±0.33 | 89.13±1.59 | 90.39±1.78 | 77.04±1.71 | 71.29±2.42 |
| RWBGE | **79.33±0.95** | 83.11±0.86 | **73.89±1.11** | 77.48±1.30 | 83.98±4.21 | 88.77±2.77 | 77.69±0.27 | 86.30±0.20 | 66.70±3.28 | 76.46±2.48 | 74.65±4.54 | 72.07±3.70 |
| PaCEr | 78.77±1.02 | **84.77±0.31** | 73.63±1.20 | **79.55±0.96** | 87.98±0.57 | 91.83±0.21 | **94.87±0.28** | **95.10±0.33** | **89.98±0.67** | 87.63±1.50 | **85.01±1.13** | **79.21±1.40** |

**Table 3: The AUC-ROC ($\pm std$) and AUC-PR ($\pm std$) of link prediction in plain networks (%).**

## 4.2 Effectiveness of PaCEr in Link Prediction

In this subsection, we evaluate the effectiveness of PaCEr in the link prediction task. The AUC-ROC and AUC-PR of PaCEr and 5 baselines on 6 datasets are presented in Table 3. First, for the AUC-ROC metric, PaCEr achieves the best AUC-ROC on 3 datasets (Ecoli, USAir and Celegans) and the second highest AUC-ROC on Cora and Citeseer. RWBGE obtains the best AUC-ROC on Cora (79.33%) and Citeseer (73.89%). Compared with RWBGE, PaCEr's AUC-ROCs on Cora and Citeseer are only 0.56% and 0.26% lower. However, on the other two datasets USAir and Celegans, PaCEr outperforms RWBGE with a 23.28% and 10.36% margin in AUC-ROC. Second, PaCEr has the best AUC-PR on 5 datasets (Cora, Citeseer, NS, Ecoli and Celegans). In detail, VGAE gets the best AUC-PR (78.00%) on Citeseer, node2vec accomplishes the best AUC-PR on Celegans (74.27%) and ARGVA reaches the best AUC-PR on Ecoli (93.31%) among all baselines. Compared with these methods, PaCEr outperforms them by 79.55% AUC-PR on Citeseer, 79.21% AUC-PR on Celegans and 95.10% AUC-PR on Ecoli. Overall, the experimental results in Table 3 demonstrate the effectiveness of PaCEr in link prediction, which shows that the positional embedding obtained by PaCEr can capture the positional information better than the embeddings produced by other baselines.

## 4.3 Effectiveness of PaCEr in Structural Node Classification

In this subsection, we demonstrate the effectiveness of PaCEr in structural node classification. As shown in Table 4, PaCEr achieves the best ACC (72.0%) on the Brazil dataset and the second highest ACC on both Europe (50.6%) and USA (64.2%). Although PaCEr's ACC is 1.8% lower than GraphWave on Europe, it has a significant ACC improvement on Brazil (7.5%) and USA (13.9%) compared with GraphWave. Similar observations can be seen in comparison with Demo-Net: Demo-Net beats PaCEr by 1.7% ACC on USA, while PaCEr outperforms Demo-Net by 10.6% on Brazil and 2.7% on Europe. The results verify the key idea of PaCEr that simply sorting the reconstructed proximity distribution matrix can lead a structural embedding with great structural expressive power.

## 4.4 Effectiveness of PaCEr-A in Attributed Node Classification on Homophilic Graphs

In this subsection, we present the effectiveness of PaCEr-A in attributed node classification on homophilic graphs. The experimental results of PaCEr-A and different methods are shown in Table 5. We can observe that PaCEr-A achieves the best accuracy on 4 datasets (i.e., Cora, Photos, CS and DBLP) and gets the second place on the Computers dataset. For the remaining two datasets Citeseer and Pubmed, it still has close performance compared with other

baselines. Specially, compared with the classical GCN and SGC, PaCEr-A outperforms them by 12% on Computers and over 6% on Photo. These observations show that utilizing the reconstructed proximity distribution matrix $\mathbf{P}$ to propagate features/attributes is effective (Eq. (5)), which can retain critical information (i.e., attribute) from distant nodes with a small weight $\mathbf{P}(u_i, u_j)$ and obtain better performance than only aggregating information from 1 or 2 hops' neighbor nodes in GCN/SGC on most datasets.

## 4.5 Effectiveness of PaCEr-H in Attributed Node Classification on Heterophilic Graphs

In this subsection, we demonstrate the performance of PaCEr-H in attributed node classification on heterophilic graphs compared with other baselines. As presented in Table 5, PaCEr-H has better performance on all three datasets than all baselines. For example, H2GCN [63] has the second best performance on the Cornell5 dataset (68.4%) and SGC [52] has the second best performance on the Squirrel dataset (37.2%). Compared to these two baselines, the proposed PaCEr-H has a 0.9% improvement on Squirrel and a 2.1% improvement on Cornell5. The superior performance of PaCEr-H can be explained from two aspects: first, the basic assumption in existing GCNs that nodes adjacent to each other are more likely to share similar attributes and labels does not hold. Consequently, GCN designed based on this assumption forces close nodes to have same labels in the classification, which in turn results in the poor performance (e.g., 27.5% ACC on the Actor dataset). Second, since the correlation between the topological information and the attribute/label information is unknown, PaCEr-H introduces an MLP module with the concatenation of the topological information and the attribute information as the input. This succinct design successfully learns the correlation and optimizes PaCEr-H's parameters with the help of nodes in the training set, which is consistent with the results in LINKX [24].

## 4.6 Ablation Study of PaCEr

In this subsection, we conduct an ablation study on PaCEr to analyze the underlying reasons for its performance improvement over baselines. The ablation study focuses on the proximity distribution matrix, since it is the central component of the proposed PaCEr. In detail, we compare the performance of adopting the reconstructed proximity distribution matrix $\mathbf{P}$ in PaCEr and directly using the proximity distribution matrix calculated by RWR ($\mathbf{R}^\top$).[8] The AUC-PRs of utilizing $\mathbf{P}$ and $\mathbf{R}^\top$ are displayed in Figure 4(a). We can find that simply using the matrix $\mathbf{R}^\top$ can already obtain good empirical

---

[8]We use $(\mathbf{R}^\top(u_i, u_j) + \mathbf{R}^\top(u_i, u_j))$ or $(\mathbf{P}(u_i, u_j) + \mathbf{P}(u_j, u_i))$ as the score for link prediction.

| Models | Brazil | Europe | USA |
|---|---|---|---|
| GraphSAGE | 40.4±3.5 | 27.2±2.2 | 31.6±2.2 |
| GCN | 43.2±6.4 | 37.1±4.6 | 43.2±2.2 |
| Union | 46.6±0.6 | 41.8±0.2 | 58.2±0.0 |
| Intersection | 45.9±0.3 | 44.3±0.2 | 57.3±0.0 |
| GAT | 38.2±12.6 | 42.4±7.3 | 58.5±2.1 |
| GraphWave | 64.5±7.9 | 52.4±5.4 | 50.3±4.0 |
| Demo-Net | 61.4±6.9 | 47.9±6.4 | 65.9±2.0 |
| PACEr | 72.0±1.8 | 50.6±1.7 | 64.2±2.2 |

Table 4: The accuracy (%) of structural node classification.

| Models | Homophilic graphs | | | | | | | Heterophilic graphs | | |
|---|---|---|---|---|---|---|---|---|---|---|
| | Cora | Citeseer | Pubmed | Computers | Photo | CS | DBLP | Squirrel | Actor | Cornell5 |
| GCN | 81.1±0.3 | 71.2±0.7 | 79.0±0.4 | 66.2±1.0 | 84.1±0.5 | 88.2±0.2 | 83.7±0.1 | 35.8±1.3 | 27.5±0.5 | 67.9±0.2 |
| SGC | 80.8±0.1 | 71.0±0.2 | 79.5±0.5 | 69.1±0.4 | 86.2±0.4 | 89.7±0.1 | 83.8±0.1 | 37.2±1.8 | 28.0±0.8 | 67.4±0.5 |
| APPNP | 82.1±0.1 | 71.8±0.1 | 79.8±0.5 | 66.7±1.1 | 83.4±1.2 | 87.8±0.1 | 83.8±0.2 | 29.5±0.9 | 32.8±0.8 | 68.3±0.5 |
| GPRGNN | 78.6±1.5 | 68.9±0.9 | 77.6±0.9 | 84.6±0.5 | 92.4±0.2 | 92.3±0.1 | 84.4±0.2 | 34.1±1.0 | 33.6±0.4 | 67.3±0.3 |
| FAGCN | 79.0±0.6 | 72.1±0.5 | 78.0±1.1 | 74.8±3.4 | 91.2±0.3 | 93.0±1.4 | 81.1±1.1 | 31.2±1.6 | 32.3±0.5 | 68.3±0.7 |
| H2GCN | 78.9±0.6 | 70.3±1.0 | 78.2±1.0 | 75.8±0.3 | 89.7±0.2 | 92.5±0.5 | 82.4±0.1 | 30.4±0.9 | 33.9±0.3 | 68.4±0.2 |
| PACEr-A/-H | 82.1±0.9 | 70.6±0.9 | 79.1±1.1 | 81.5±0.7 | 92.9±0.1 | 93.6±0.2 | 84.6±0.5 | 38.1±0.9 | 34.5±1.5 | 70.5±0.4 |

Table 5: The accuracy (±std) of node classification on homophilic graphs and heterophilic graphs (%).

results such as 94.77% AUC-PR on the Ecoli dataset, which are better than all baselines in Table 3. Through matrix factorization, using the reconstructed proximity distribution matrix $\mathbf{P}$ can capture the positional information better, and have further performance gains over $\mathbf{R}^\top$ (e.g., 3.83% AUC-PR on Citeseer). For the structural node classification task (Figure 4(b)), $R(\mathbf{R}^\top)$ also performs well, which verifies Proposition 1 that $R(\mathbf{R}^\top)(u_i,:)$ maintains a good structural expressive power. PACEr further boosts the accuracy of the structural node classification by about 8% on the Brazil dataset, which demonstrates that compared with $R(\mathbf{R}^\top)(u_i,:)$, PACEr can capture structural information better.

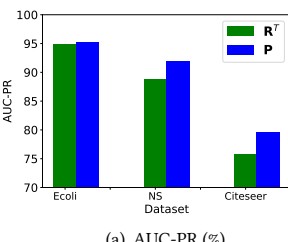 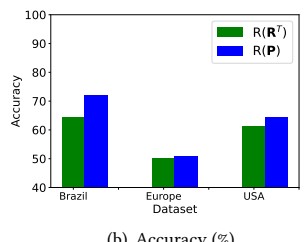

(a) AUC-PR (%)     (b) Accuracy (%)

Figure 4: Ablation study on $\mathbf{P}$ *vs.* $\mathbf{R}^\top$ in link prediction (a), and $R(\mathbf{P})$ *vs.* $R(\mathbf{R}^\top)$ in structural node classification (b).

## 5 RELATED WORKS

**Network Embedding.** Network embedding maps nodes, subgraphs or the entire graph to low dimensional vectors. It can be traced back to matrix factorization [25]. Since Word2Vec [26] emerges, Deep-Walk [32] and node2vec [14] embrace a similar idea of conducting a random walk on the graph to learn the node representations in homogeneous networks. LINE [42] owns similar positional embedding module to our proposed PACEr, but it merely considers first-order and second-order proximity and does not possess a structural embedding module. For heterogeneous networks or attributed networks, metapath2vec [8] and DANE [12] integrate different types of nodes or attributes into the random walk. Recently, graph convolutional network (GCN) [19] has turned out to be a powerful framework for graph representation learning and many GCN-based methods have been proposed. For example, graph attention network (GAT) [46] introduces the attention mechanism into GCN and simple graph convolution (SGC) [52] removes the non-linear layer in GCN. Geom-GCN [31], H2GCN [63], GPRGNN [6] and FAGCN [3] are proposed to handle heterophilic graphs. More advanced GCNs including APPNP [21] and GraphSAGE [15] can be found in the survey by Wu et al. [54].

**Positional Embedding vs. Structural Embedding.** On the one hand, struct2vec [34] notices that previous methods such as Deep-Walk [32] and node2vec [14] tend to make the embeddings of connected nodes similar. However, nodes with identical local structures obtain different embeddings in DeepWalk and node2vec. On the other hand, PGNN [59] points out that classical GCNs such as GCN [19] can not distinguish isomorphic node pairs and capture the positional information. GIN [55] proves that classical GCNs are *at most* as powerful as the Weisfeiler-Lehman (WL) test [51]. Most works on positional embedding and structural embedding focus on improving the expressive power of classical GCN and integrating positional information. One direction to fulfill this goal is to utilize augmented node features to make isomorphic nodes become position-aware and distinguishable. For instance, GAE [20] sets one-hot feature for each node, SEAL [60] and ID-GNN [58] embrace identity (position)-aware labelling tricks. Another direction is to conduct positional encoding. For instance, PGNN [59] builds anchor node sets and learns a distance-weighted aggregation scheme. Srinivasan et al. [41] propose that the eigenvectors of graph Laplacian matrix can encode the positional information. PEG [48] further aggregates the original node features and the positional features earned by eigen-decomposition, DeepWalk and node2vec in different channels. For the relation between positional embedding and structural embedding, Srinivasan et al. [41] prove that positional embedding cannot capture more information than *multi-node set* structural embedding. It is important to note that the result in [41] does not contradict with our findings in this paper because we focus on *single-node level* positional embedding and structural embedding. Additional related works can be found in [62].

## 6 CONCLUSION

In this paper, we investigate the relation between positional embedding and structural embedding in network embedding. Specifically, we find that with the help of the intermediate proximity distribution matrix, we can acquire the *corresponding* structural embedding by performing direct transformations on the positional embedding. Based on this finding, we propose a novel network embedding algorithm PACEr, which adopts the RWR proximity distribution matrix to optimize the positional embedding. Then, the structural embedding is learned by sorting the reconstructed proximity distribution. Furthermore, for attribute graphs, we propose two variants of PACEr, including PACEr-A and PACEr-H for homophilic and heterophilic graphs, respectively. Through extensive empirical evaluations on four graph learning tasks, we corroborate the effectiveness of PACEr.

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

# 7 APPENDIX

In the section, we include the proofs of propositions and the reproducibility of this work.

## 7.1 Proofs

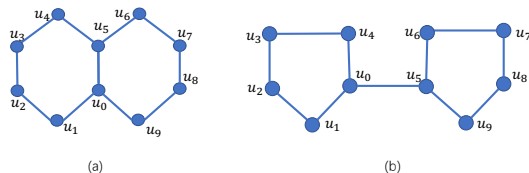

**Figure 5: Two non-isomorphic graphs in Sato et al. [37] that can not be discriminated by 1-WL test.**

PROPOSITION 1. $R(\mathbf{R}^\top(u_i, :))$ is permutation invariant and its expressive power is not worse than the expressive power of 1-WL test.

PROOF. We first prove that $R(\mathbf{R}^\top(u_i, :))$ is permutation invariant. To prove this, we only need to prove $R(\mathbf{R}^\top(u_i, :)) = R(\mathbf{R}^\top(\pi(u_i), :))$. First, because the graph topology and the actual starting node ($u_i$ or $\pi(u_i)$) has not changed in the graph under a node index permutation, for any value appearing in the vector $\mathbf{R}^\top(u_i, :)$, it will also appear in the vector $\mathbf{R}^\top(\pi(u_i), :)$ and identical values will have same times of appearances, and vice versa. Then, we prove $R(\mathbf{R}^\top(u_i, :)) = R(\mathbf{R}^\top(\pi(u_i), :))$ by contradiction. We sort both $\mathbf{R}^\top(u_i, :)$ and $\mathbf{R}^\top(\pi(u_i), :)$ in a descending order. Assume $\mathbf{R}^\top(u_i, :) \neq \mathbf{R}^\top(\pi(u_i), :)$ and the first unequal value has index $j$, which means that $R(\mathbf{R}^\top(u_i, :))_j \neq R(\mathbf{R}^\top(\pi(u_i), :))_j$. Without loss of generality, we can assume that $R(\mathbf{R}^\top(u_i, :))_j > R(\mathbf{R}^\top(\pi(u_i), :))_j$. Due to the descending order, $R(\mathbf{R}^\top(u_i, :))_j > R(\mathbf{R}^\top(\pi(u_i), :))_k$ for any $k > j$. This means that $R(\mathbf{R}^\top(u_i, :))_j$ has no corresponding value in $\mathbf{R}^\top(\pi(u_i), :)$ and this leads to the contradiction. To prove that its expressive power is not worse than the expressive power of 1-WL test, following Dwivedi et al. [10], we only need to find a graph pair that are non-isomorphic and can not be discriminated by the 1-WL test, but can be distinguished by $R(\mathbf{R}^\top(u_i, :))$. We adopt the examples in Sato et al. [37] (shown in Figure 5). These two graphs are non-isomorphic and have been proved by [37] that can not be discriminated by 1-WL test. Now, we calculate the $R(\mathbf{R}^\top(u_i, :))$ for all nodes in both graphs. $R(\mathbf{R}^\top(u_i, :))$s for nodes in Figure 5(a) and $R(\mathbf{R}^\top(u_j, :))$s for nodes in Figure 5(b) are as the following with $10^{-2}$ base:

$$\begin{bmatrix} 27.96 & 13.07 & 10.89 & 10.89 & 6.98 & 6.98 & 6.06 & 6.06 & 5.54 & 5.54 \\ 25.95 & 16.33 & 14.87 & 9.09 & 9.05 & 6.42 & 6.42 & 4.23 & 4.07 & 3.53 \\ 28.01 & 15.75 & 14.87 & 10.47 & 9.05 & 8.31 & 4.23 & 3.53 & 2.97 & 2.76 \\ 28.01 & 15.75 & 14.87 & 10.47 & 9.05 & 8.31 & 4.23 & 3.53 & 2.97 & 2.76 \\ 25.95 & 16.33 & 14.87 & 9.09 & 9.05 & 6.42 & 6.42 & 4.23 & 4.07 & 3.53 \\ 27.96 & 13.07 & 10.89 & 10.89 & 6.98 & 6.98 & 6.06 & 6.06 & 5.54 & 5.54 \\ 25.95 & 16.33 & 14.87 & 9.09 & 9.05 & 6.42 & 6.42 & 4.23 & 4.07 & 3.53 \\ 28.01 & 15.75 & 14.87 & 10.47 & 9.05 & 8.31 & 4.23 & 3.53 & 2.97 & 2.76 \\ 28.01 & 15.75 & 14.87 & 10.47 & 9.05 & 8.31 & 4.23 & 3.53 & 2.97 & 2.76 \\ 25.95 & 16.33 & 14.87 & 9.09 & 9.05 & 6.42 & 6.42 & 4.23 & 4.07 & 3.53 \end{bmatrix} \quad (7)$$

$$\begin{bmatrix} 28.56 & 12.47 & 11.79 & 11.79 & 8.72 & 8.72 & 5.15 & 5.15 & 3.80 & 3.80 \\ 26.83 & 17.69 & 16.04 & 10.92 & 9.65 & 7.72 & 3.19 & 3.19 & 2.35 & 2.35 \\ 29.03 & 16.98 & 16.04 & 13.08 & 10.92 & 5.71 & 2.35 & 2.35 & 1.74 & 1.74 \\ 29.03 & 16.98 & 16.04 & 13.08 & 10.92 & 5.71 & 2.35 & 2.35 & 1.74 & 1.74 \\ 26.83 & 17.69 & 16.04 & 10.92 & 9.65 & 7.72 & 3.19 & 3.19 & 2.35 & 2.35 \\ 28.56 & 12.47 & 11.79 & 11.79 & 8.72 & 8.72 & 5.15 & 5.15 & 3.80 & 3.80 \\ 26.83 & 17.69 & 16.04 & 10.92 & 9.65 & 7.72 & 3.19 & 3.19 & 2.35 & 2.35 \\ 29.03 & 16.98 & 16.04 & 13.08 & 10.92 & 5.71 & 2.35 & 2.35 & 1.74 & 1.74 \\ 29.03 & 16.98 & 16.04 & 13.08 & 10.92 & 5.71 & 2.35 & 2.35 & 1.74 & 1.74 \\ 26.83 & 17.69 & 16.04 & 10.92 & 9.65 & 7.72 & 3.19 & 3.19 & 2.35 & 2.35 \end{bmatrix} \quad (8)$$

□

We can observe that $\forall u_i$ in Figure 5 (a) and $\forall u_j$ in Figure 5 (b), $R(\mathbf{R}^\top(u_i, :)) \neq R(\mathbf{R}^\top(u_j, :))$. Therefore, $R(\mathbf{R}^\top(u_i, :))$ can distinguish these two graphs.

## 7.2 Reproducibility

**Datasets.** (1) For the link prediction task, we use 6 datasets: NS [27], Ecoli [40], USAir [47], Celegans [50], Citeseer [57][9], and Cora [57]. We randomly split edges in every dataset into 70/10/20% for training, validation and test. Since the link prediction task also needs negative edges, we randomly sample same amount of non-existent edges for validation and test. (2) For the structural node classification task, we use 3 benchmark airport datasets, including Brazil, Europe and USA [34], which are commonly used in the evaluation of structural embedding [18, 34, 53]. We follow the split in DEMO-Net [53] with 33/33/34% nodes for training, validation, and test. (3) For the homophilic node classification task, we use 7 benchmark datasets: Cora [57], Citeseer [57], Pubmed [57], Computers [38], Photo [38], CS [38] and DBLP [4]. For these 3 datasets (Cora, Citeseer, Pubmed), we follow the standard split in the GCN [19] paper. For the remaining 4 datasets, we randomly split the them into 20/20/60% for training, validation and test. (4) For the heterophilic node classification task, we use three datasets: Squirrel [36], Actor [31], Cornell5 [24] and also adopt a 20/20/60% training/validation/test split.

---

[9]The Citeseer dataset used in the link prediction task does not contain attributes.

