# OpenReview forum: "PACER: Network Embedding From Positional to Structural"
_ACM.org/TheWebConf/2024/Conference — TheWebConf24_

### Official Review · Reviewer_zDfS · 2023-10-24

**Novelty:** 6
**Technical Quality:** 6

**Review:**

Thanks for submitting the work to the Web Conference.

Positional and structural embeddings are both very important in web mining tasks in practice, such as fraud detection, link prediction, etc. In addition, the authors attempt to unify both types of embeddings in one analytical framework, which is of interest to the audience of the Web, as previously these two types of embeddings seem to be independent and even contradictory to each other. From this perspective, the problem and the motivation of this paper are both clear. I also appreciate that the paper is in general well written and easy to follow even for non-experts, as sufficient technical intuitions are provided.

The technical solution of this paper is intuitively simple but makes good sense. I appreciate the observation that sorting the proximity distribution provides insights towards structural embeddings, which is simple but insightful. The introduction of random walk with restart, although is widely used to depict general proximity, is also technically sound. Moreover, theoretical evidence is given to link the sorted RWR matrix with the WL-test. Efforts to extend factorization-based solutions to attributed networks are also appreciated.

Experiments are extensive, covering different types of node learning tasks (link prediction, structural classification, etc). PACER is competitive in all types of tasks, showing its effectiveness in unifying structural node embeddings and positional ones.

Overall I think it is a good work with important problem, good motivation, simple but insightful solutions, and extensive experiments. My questions to the authors are given in the 'Questions' section.

**Questions:**

Q1, I think the authors can better discuss the difference between PACER and [41], which I think is the most relevant work related to this one. Specifically, how are 'multi-node set' structural embeddings different from single-node embeddings?

Q2: Although I appreciate the overall motivation and technical quality of this paper, I have a slight concern regarding baseline selections of this paper. Specifically, I was wondering whether it is possible to ensemble a structural node embedding method with a positional one (e.g. concatenating their outputs), and what is its performance. It is very straightforward but may be extremely effective, so I would like to see such a comparison.

Q3: It seems strange that PACER-A generates structural embeddings from node features and the reconstructed proximity distribution. Specifically, in homophilous graphs, connected nodes may share similar node features. Thus, the node features $\mathbf{X}$ are somehow inherently positional. In addition, the $P$ is also positional (as it encodes proximity). Therefore, why the combination of these two becomes 'structural embeddings' is not very clear.

Q4: I would like to see some additional ablation studies to better showcase the insights of PACER. First, it would be good if the authors can compare using $R^T$ and $D^{-1}A$, as this marks the difference between ordinary GNN/GAE and the proposed PACER. Second, maybe the authors can provide some analysis on the restart probability. Intuitively, a larger restart probability will lead to embeddings that are 'local', while a smaller one leads to 'global' embeddings that may be effective for heterophilous graphs. It would be good to provide such analyses.

Q5: Any insights why PACER-A performs not so good in Homophilic graphs?

**Ethics Review Description:**

Not required.

**Reviewer Confidence:**

3: The reviewer is confident but not certain that the evaluation is correct

**Scope:**

4: The work is relevant to the Web and to the track, and is of broad interest to the community

---

### Official Review · Reviewer_nkHX · 2023-11-22

**Novelty:** 4
**Technical Quality:** 5

**Review:**

This paper focuses on incorporating the relationship between structural embedding and positional embedding via proximity distribution. The authors propose three models for non-attribute, homophilic, and heterophilic graphs. The above models outperform the selected baselines. However, there are some weaknesses that need to be further improved.

1. The details of the model are missing. The proposed models seem to be un/self-supervised ones. However, how to optimize MLP in PRCER-H is confusing.

2. Missing important baselines. There are some key baselines of both PE and SE (e.g., methods in Table 1). Moreover, baselines need to be consistent regardless of task (i.e., link prediction and node classification). In addition, some heterophilic graph learning methods should also be included in Table 5.

**Questions:**

1. Why KL-divergence? Why not the traditional pair-wise objectives in DeepWalk?
2. How about the heterophilic graphs without attributes?

**Reviewer Confidence:**

3: The reviewer is confident but not certain that the evaluation is correct

**Scope:**

3: The work is somewhat relevant to the Web and to the track, and is of narrow interest to a sub-community

---

### Official Review · Reviewer_zXAw · 2023-11-27

**Novelty:** 3
**Technical Quality:** 4

**Review:**

The paper introduces a novel Network Embedding pipeline, which tries to capture both positional and structural similarity between nodes in a graph. The main idea is to manipulate a positional embedding based on RWR proximity matrix, by essentially sorting rows in the reconstructed proximity distribution and comparing the resulting vectors.

*** More detailed comments:

line 43: 'and have achieved
strong empirical performance' -> strong performance for which tasks? Add these tasks

line 144: 'redundant given informative positional embedding' -> rewrite sentence

line 219: I have never seen this way of writing a graph as G=(V,E,A). No need to add the adjacency matrix A to that triplet. It is redundant information, it follows from the edges E. Instead, create a new sentence simply stating that 'A is the corresponding adjacency matrix of G', or something similar.

line 239: Specify the domain and range of the functions f, and also f^k.

Definition 2: This is a lot of notation for a simple concept as a k-hop subgraph. Try to simplify the definitions, using more words to descibe the objects.

Definition 5 (line 258) is written in a rather unclear manner. It 'can be denoted' -> Why 'can'? Are you denoting it or not? Do not leave room for such ambiguity in a definition. Also, first define phi, l, and W before introducing g.

line 318: Move definition of row-normalized adj. matrix to the beginning of Sect. 2.

line 364: Use 'testing' instead of 'testifying'.

***EDIT: Thank you for the answer and the additional experiments on the extra datasets in the EvalNE benchmark evaluation framework.
I have raised my score.

**Questions:**

line 416: Instead of RWR matrix, have you considered simply using higher-order proximity matrices of the adjacency matrix? These are matrices that can be expressed as a polynomial of A, with positive coefficients.
One can directly use fast top-d eigendecomposition of the adj. matrix A (since A is typically sparse), to obtain the top-d eigendecomposition of these polynomials, and thus any rank-d approximations. Such polynomials of degree k contain a lot of information of k-hop neighborhoods. See the following paper for exmaple; Z. Zhang, P. Cui, X. Wang, J. Pei, X. Yao, and W. Zhu, “Arbitrary-order
proximity preserved network embedding,” in KDD, 2018.

The empirical evaluation/contribution is very limited in my opinion.
First of all, the datasets are small in size. This is a serious limitation compared to other state-of-the-art embedding methods.
Secondly, there is not enough comparison with other existing NE methods, in particular for the task of link prediction.
For example, it is known that node2vec is outperformed by almost all subsequent NE methods, so there is no real need to compare with node2vec. Instead, try to compare with other state-of-the-art methods.

The paper by Mara et. al in DSAA 2020 (Benchmarking Network Embedding Models for
Link Prediction: Are We Making Progress?) shows that very simple heuristics for link prediction (such as the number of common neighbors between two nodes) often give very good results. Did you compare with these simple heuristics? Also, those authors have a benchmark evaluation framework (EvalNE) for evaluating network embedding methods on several tasks. Consider using those.

**Ethics Review Description:**

no issues

**Reviewer Confidence:**

3: The reviewer is confident but not certain that the evaluation is correct

**Scope:**

4: The work is relevant to the Web and to the track, and is of broad interest to the community

---

### Decision · Program_Chairs · 2024-01-22

**Decision:**

Accept

**Comment:**

This paper studies the relations between structural and positional similarities in the nodes of a graph.
 Many aspects of the paper were appreciated by all reviewers (interesting and sound motivation, extensive experiments etc.)
 Some concerns were raised especially about some confusing details in the description of the model and the fact that
 some important baselines are missing.
 The authors did a very good job in answering most of the negative comments, though. Almost all of the concerns seem
 to be solved, albeit the authors are warmly invited to revise their work including at least the most important comments in
 the final version.